 SHORT REPORT

# Improving emotional-action control by targeting long-range phase-amplitude neuronal coupling

**Bob Bramson[1]\*, Hanneke EM den Ouden[1], Ivan Toni[1†], Karin Roelofs[1,2†]**

[1]Donders Institute for Brain, Cognition and Behaviour, Centre for Cognitive Neuroimaging, Radboud University Nijmegen, Nijmegen, Netherlands; [2]Behavioural Science Institute (BSI), Radboud University Nijmegen, Nijmegen, Netherlands

**Abstract** Control over emotional action tendencies is essential for everyday interactions. This cognitive function fails occasionally during socially challenging situations, and systematically in social psychopathologies. We delivered dual-site phase-coupled brain stimulation to facilitate theta-gamma phase-amplitude coupling between frontal regions known to implement that form of control, while neuropsychologically healthy human male participants were challenged to control their automatic action tendencies in a social–emotional approach/avoidance-task. Participants had increased control over their emotional action tendencies, depending on the relative phase and dose of the intervention. Concurrently measured fMRI effects of task and stimulation indicated that the intervention improved control by increasing the efficacy of anterior prefrontal inhibition over the sensorimotor cortex. This enhancement of emotional action control provides causal evidence for phase-amplitude coupling mechanisms guiding action selection during emotional-action control. Generally, the finding illustrates the potential of physiologically-grounded interventions aimed at reducing neural noise in cerebral circuits where communication relies on phase-amplitude coupling.

**\*For correspondence:**
b.bramson@donders.ru.nl

†These authors contributed equally to this work

**Competing interests:** The authors declare that no competing interests exist.

## Introduction

The ability to control emotional actions is paramount for successful engagement in human social interactions (*Hare, 2017*). Long-standing theorizing and accumulating empirical evidence indicate that affective cues automatically activate approach-avoidance action tendencies (*Ridderinkhof, 2017*; *Phaf et al., 2014*). Effective emotion control requires the cognitive capacity to suppress those automatic action tendencies and to select an alternative course of action (*Etkin et al., 2006*; *Koch et al., 2018*). The importance of emotional-action control becomes apparent when it is disrupted: In social psychopathologies such as social anxiety, the inability to override social avoidance tendencies constitutes the core maintaining factor of the disorder (*Craske and Stein, 2016*). There is great interest in potentiating this cognitive capacity to enhance treatment efficacy, as well as to help professionals exposed to socially challenging situations. However, improving human emotional-action control has proven difficult (*den Uyl et al., 2015*). In this study, we use a brain stimulation intervention designed to enhance synchrony within a cerebral circuit known to support emotional-action control (*Bramson et al., 2018*; *Volman et al., 2011a*). By modeling how effective and structural connectivity of that cerebral circuit mediates the behavioral effects of the intervention, we provide an account of its neural effects, paving the way for physiologically-grounded therapeutic interventions in social-emotional disorders (*Voytek and Knight, 2015*).

Previous non-invasive brain stimulation interventions have been successful in *reducing* emotional-action control. This was achieved by disrupting neural activity – putatively by injecting neural noise (*Schwarzkopf et al., 2011*) – in a region known to coordinate emotional-action control: the anterior prefrontal cortex (aPFC) (*Volman et al., 2011a*). Here we explore whether it is possible to *enhance*

emotional-action control by using brain stimulation aimed at reducing neural noise, targeting a corti-cal circuit known to regulate those action-tendencies (*den Uyl et al., 2015*; *Bramson et al., 2018*; *Volman et al., 2011a*). This gain-of-function intervention is grounded on recent insights showing that emotional-action control requires neural synchronization between aPFC theta-band rhythm and sensorimotor broadband gamma activity (*Bramson et al., 2018*; *Voytek et al., 2015*; *Canolty and Knight, 2010*). We reasoned that endogenous neural synchronization might be enhanced by apply-ing separate time-varying electric fields (transcranial alternating current stimulation; tACS [*Fröhlich and McCormick, 2010*]) to aPFC and sensorimotor cortex (SMC). tACS influences spike timing of individual neurons (*Fröhlich and McCormick, 2010*; *Krause et al., 2019*; *Johnson et al., 2020*), and entrains neural rhythms to its frequency and phase (*Reinhart and Nguyen, 2019*; *Polanía et al., 2015*). We applied dual-site phase-coupled tACS to enhance the endogenous syn-chronization of SMC gamma-band power (75 Hz) with the peaks of aPFC theta-band rhythm (6 Hz) evoked during emotional-action control (*Bramson et al., 2018*; *Nowak et al., 2017*; *Akkad et al., 2020*). Enhanced phase-amplitude coupling would reduce neural noise in aPFC-SMC communication (*Voytek and Knight, 2015*; *Canolty and Knight, 2010*), allowing for improved control over emo-tional-action tendencies. Crucially, we apply this aPFC-SMC stimulation while 41 human male partici-pants with no history of mental illness perform an emotional-action control task (*Figure 1A*). We compare the online behavioral and neural effects of in-phase, anti-phase, and sham couplings between the power envelope of SMC gamma stimulation and the peaks of aPFC theta stimulation (*Figure 2A*,B). Importantly, concurrent whole-brain BOLD-fMRI quantified local and remote dose-dependent cerebral effects of the electrical stimulations (*Evans et al., 2020*). tACS effects on each participants' cerebral connectivity were further qualified using dynamic causal modeling (*Friston et al., 2003*), informed by MR-tractography (*Behrens et al., 2007*).

## Results

We manipulated emotion-action control through a social-emotional approach-avoidance task, where human male participants use a joystick to rapidly approach or avoid happy or angry faces (*Figure 1A*). People tend to approach happy faces and avoid angry faces (*Phaf et al., 2014*). Over-riding these *affect-congruent* action-tendencies and instead generating *affect-incongruent* actions (approach-angry and avoid-happy) requires control, which is implemented through the aPFC, parie-tal/SMC, and amygdala/hippocampal regions (*Bramson et al., 2018*; *Volman et al., 2011a*). Accord-ingly, in the sham condition of this study, participants' error rates and aPFC BOLD activity increased when incongruent approach-avoidance responses are compared to affect-congruent responses, reproducing previous effect sizes (*Bramson et al., 2018*; *Volman et al., 2011a*) – *Figure 1B,C*.

We quantified the magnitude of the physiological effects of tACS on the underlying neural tissue through concurrent tACS-fMRI. We used BOLD effects of theta-band tACS stimulation over aPFC (in-phase + anti-phase stimulation epochs versus sham; *Figure 2D*), as a dose-dependent metric of tACS effects (tACS-dose; *Evans et al., 2020*), independent of task performance. In line with our expecta-tions, higher tACS-dose on aPFC increased the emotion-control enhancement induced by in-phase stimulation (in-phase aPFC-theta/SMC-gamma tACS) [emotional control*phase condition*tACS-dose interaction: *Figure 2E*]. Interestingly, across participants, the local cortical (BOLD) effects of the inhibi-tory theta-band aPFC stimulation (*Scheeringa et al., 2011*) correlated positively with improved emo-tion-control (indexed by decreases in behavioral congruency-effects during the in-phase condition but not in the anti-phase condition –*Figure 2D*). These dose- and phase-dependent results, predicted on the knowledge that emotional-action control requires synchronization between specific endogenous rhythms (*Bramson et al., 2018*; *Helfrich and Knight, 2016*), indicate that in-phase aPFC-theta/SMC-gamma tACS increases participants' control over their emotional action-tendencies.

The observed cognitive benefits of the tACS intervention could arise from direct modulation of aPFC-SMC connectivity, or be mediated by other regions (*Figure 3A*). We arbitrated between those possibilities using two complementary approaches. First, we tested whether the amygdala mediates these effects. This region is a prime candidate as it is connected to both aPFC and SMC, and is strongly involved in emotional-action control (*Volman et al., 2011a*; *Etkin et al., 2015*). We distin-guished between a number of anatomically plausible circuit-level effects of the tACS intervention using dynamic causal modeling (*Friston et al., 2003*) on regional BOLD timeseries in the amygdala, aPFC and SMC, measured during performance of the approach-avoidance task. Model comparison

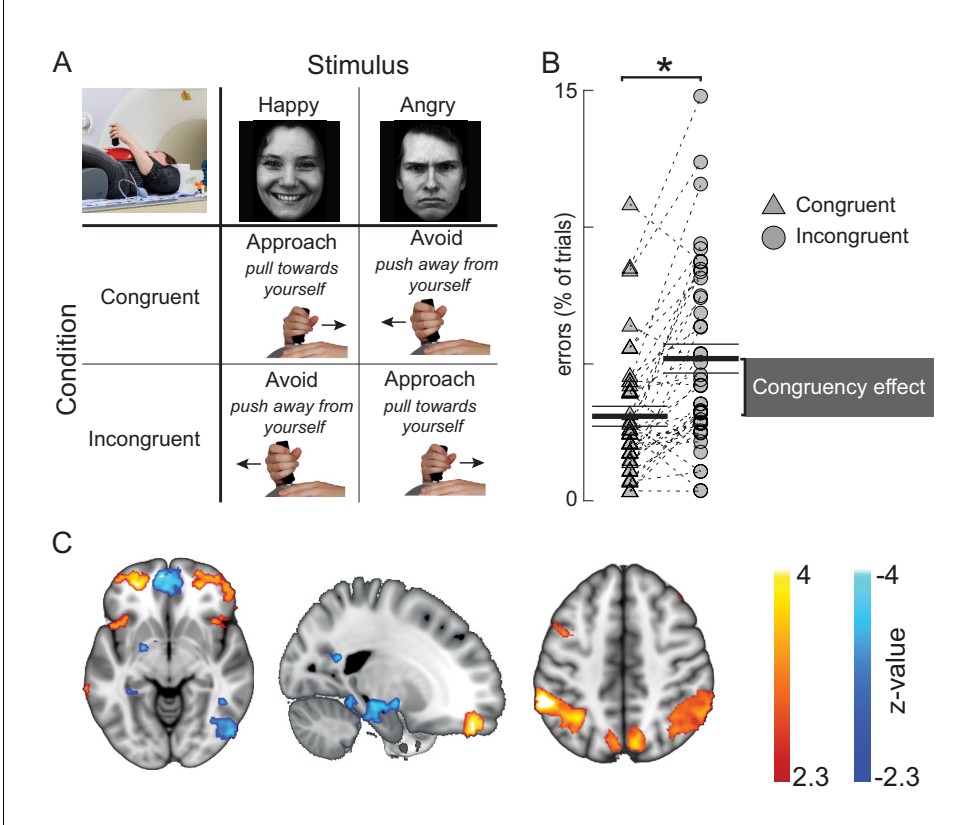

**Figure 1.** Behavioral and cerebral effects of the approach-avoidance task used to manipulate control over emotional action tendencies. (**A**) Conceptual visualization of the approach-avoidance task. Participants pushed- or pulled a joystick away- or toward themselves to approach- or avoid happy and angry faces. Approaching angry- and avoiding happy faces is incongruent with action tendencies to approach appetitive and avoid aversive situations. (**B**) Behavioral results in the sham condition of the task. Participants make more errors in the incongruent trials (circles) than in the congruent trials (triangles). Black lines visualize the mean and standard error of the mean. Gray bar depicts the group average congruency effect. (**C**) Approach-avoidance congruency-related BOLD changes (p<0.01 cluster-level inferences corrected for multiple comparisons). Trials involving responses incongruent with automatic action tendencies showed stronger BOLD signal in anterior prefrontal- and parietal areas, and reduced signal in the left amygdala/hippocampus and medial prefrontal cortex.

supports a circuit where tACS modulates aPFC→SMC, aPFC→amygdala, and amygdala→SMC connections (*Figure 3—figure supplement 1*). Crucially, the effect of stimulation on emotional-action control was driven by tACS modulation of a specific component of that circuit, the task-related aPFC→SMC connectivity (emotional-control*phase-condition*aPFC→SMC effective connectivity; *Figure 3*). *Figure 3* visualizes this behavioral- and connectivity-related difference in tACS effects, contrasting the behavioral effects evoked in participants with strong or weak in-phase tACS modulation of aPFC→SMC connectivity. Emotional control was enhanced in those participants with strong in-phase tACS modulation of aPFC-SMC connectivity: the congruency effect observed in the anti-phase and sham conditions disappeared during the in-phase condition. By contrast, emotional control remained unaffected in those participants without in-phase tACS modulation of aPFC→SMC connectivity: there were similar congruency effects across in-phase, anti-phase, and sham conditions. Second, we assessed whether inter-participant variance in the circuit-level effect of the tACS intervention can be understood in terms of inter-participant variance in structural connectivity between aPFC and SMC. In support of this notion, participants with higher fractional anisotropy (FA; a measure of white matter integrity) in the white matter beneath BA6 had stronger inhibitory coupling between aPFC and SMC during in-phase tACS (as compared to anti-phase tACS) (supplementary results).

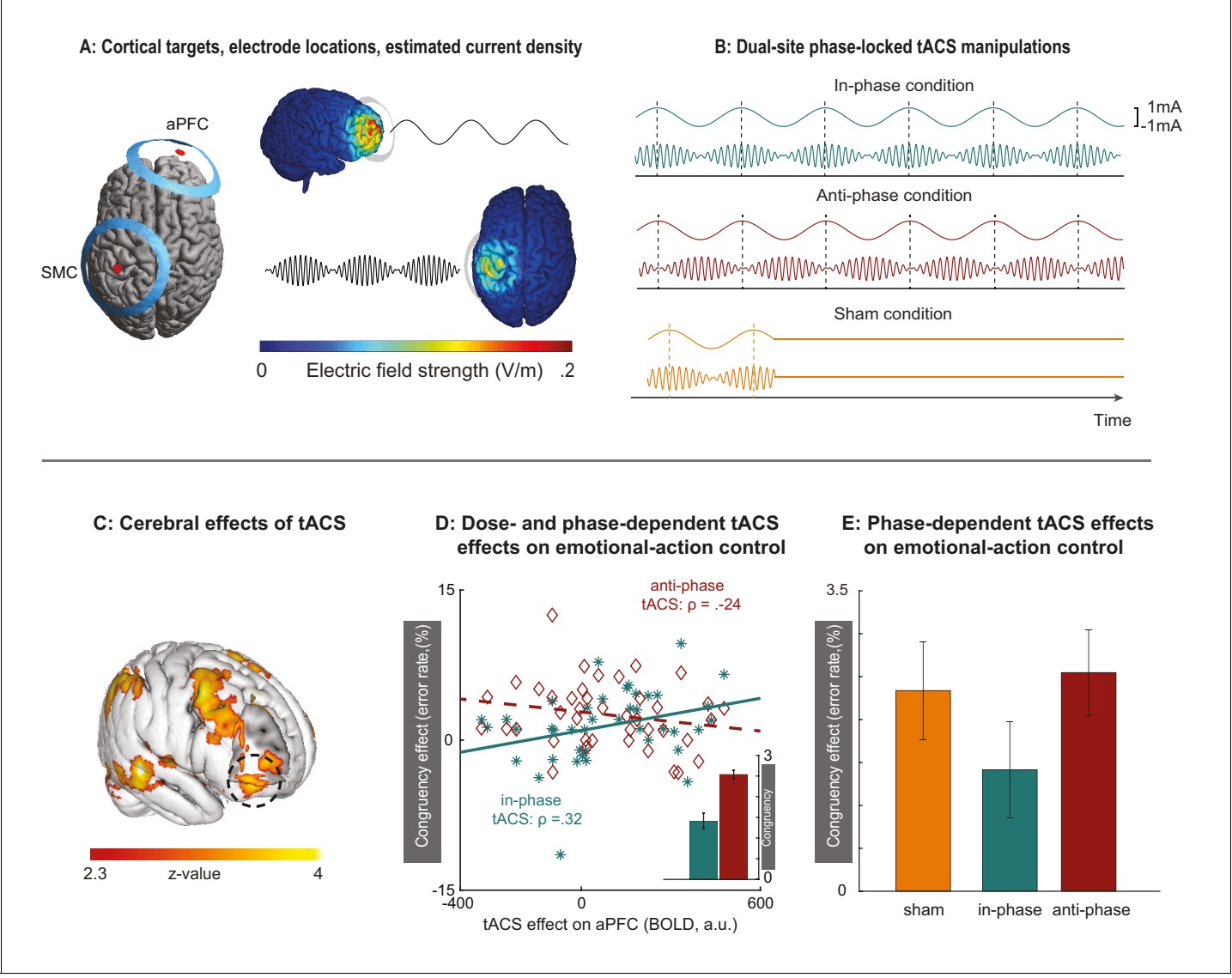

**Figure 2.** Behavioral effects of dual-site phase-coupled tACS on emotional-action control are dose- and phase-dependent. (**A**)Two sets of ring electrodes were placed over the right aPFC and left SMC. Modeling of the current density showed that stimulation reached both regions of interest with intensities known to support phase entrainment when matched to the endogenous rhythms (*Reato et al., 2010*). (**B**) During the experiment, stimulation conditions were alternated between in-phase, anti-phase, and sham conditions in pseudo-random fashion. The 75 Hz stimulation over SMC was amplitude-modulated according to the 6 Hz stimulation over aPFC, either in-phase or anti-phase with the peaks of the 6 Hz aPFC stimulation. Sham consisted of an initial stimulation of 10 s, after which stimulation was terminated. (**C**) Concurrent tACS-fMRI quantified changes in BOLD signal evoked by the tACS intervention, across in-phase and anti-phase conditions, and independently from task performance. Online physiological effects of tACS are evident both under the aPFC electrode (black circle) and in other cortical regions. (**D**) Participants with stronger inhibitory responses to theta-band stimulation over aPFC (manifested as decreased BOLD signal; *Scheeringa et al., 2011*) improved their control over emotional actions (decreased congruency effect) during aPFC-SMC in-phase tACS (in green) but not during aPFC-SMC anti-phase tACS (in red), reflected in the interaction between emotional control (congruent, incongruent), stimulations phase (in-phase, anti-phase) and tACS-dose (BOLD signal in aPFC during stimulation versus sham [panel C]); $F(1,39) = 9.3$, $p=0.004$, partial $eta^2 = 0.19$. Inset bar graphs illustrate group average parameter estimates of the congruency effects corrected for tACS-dose. The direction of the effect in the anti-phase condition tentatively suggests that stronger entrainment to the theta-band stimulation (decreased BOLD) increases congruency effects, as would be expected when aPFC-SMC communication is disrupted. (**E**) Without controlling for inter-participant variability in tACS-dose, the differential phase effect on emotional-actions control is less statistically reliable ($p=0.06$, partial $eta^2 = 0.088$).

The online version of this article includes the following figure supplement(s) for figure 2:

**Figure supplement 1.** Behavioral and neural effects in different task conditions.

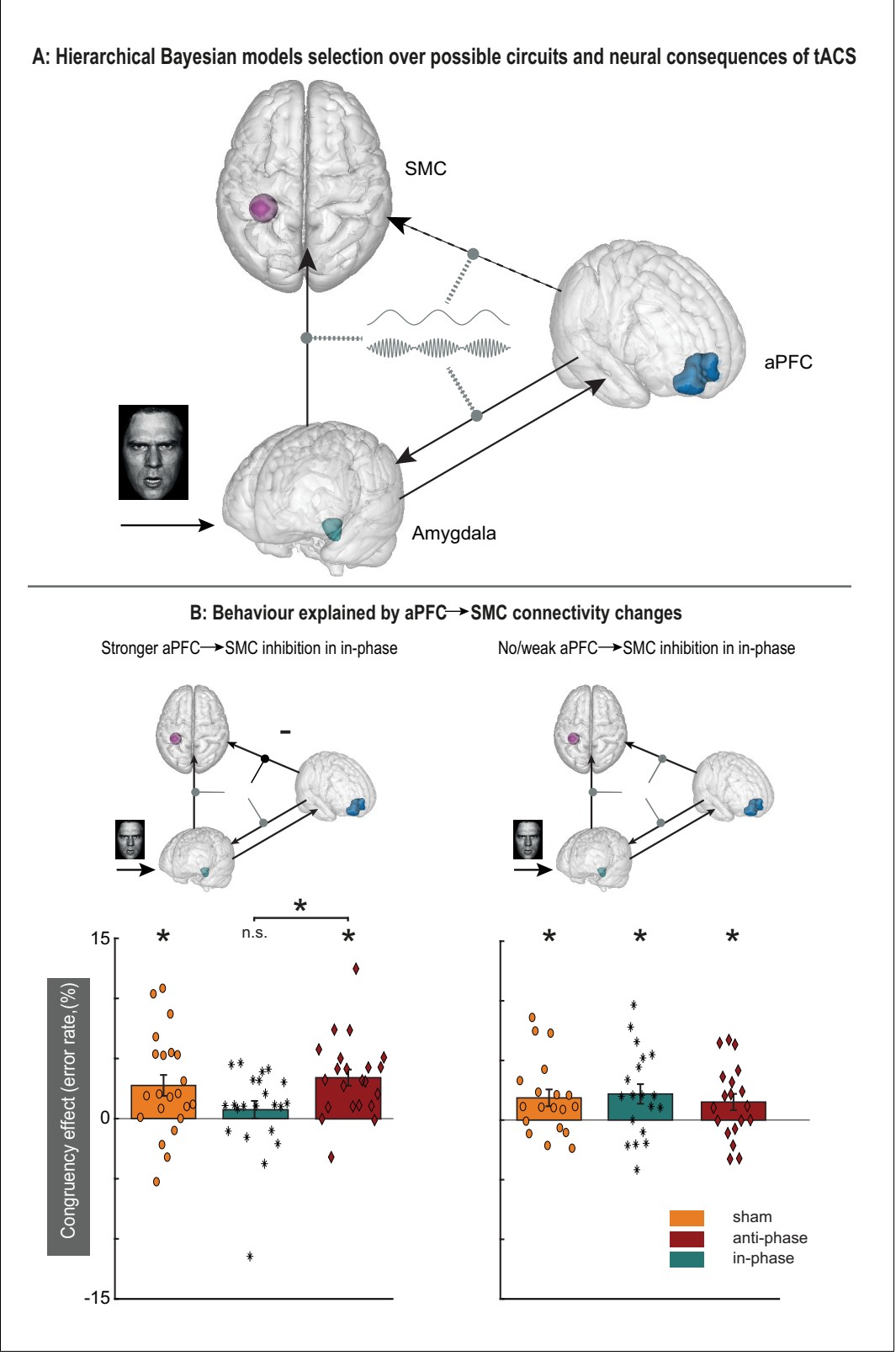

**Figure 3.** Modulatory effects of dual-site phase-coupled tACS on emotional-action control depend on effective connectivity between aPFC and SMC. (**A**) Model selection compared models with and without a direct connection between aPFC→SMC (dashed arrow), and tACS modulations on different connections (gray dashed oval arrows). (**B**) Model selection indicates that tACS affects multiple connections in the network (top panels; supplementary

*Figure 3 continued on next page*

*Figure 3 continued*

materials), but only tACS-related changes in connectivity between aPFC→SMC predict behavioral effects of the stimulation (interaction between emotional control [congruent, incongruent], stimulations phase [in-phase, anti-phase] and p=0.019, np$^2$ = 0.13; lower panels). Those participants with stronger inhibitory influence of aPFC over SMC in the in-phase condition (lower left panel, n = 22) showed decreases in congruency effects in the in-phase condition but not in the anti-phase and sham condition. The rest of the participants did not show a differential effect between stimulation conditions. Asterisks: p<0.01.

The online version of this article includes the following figure supplement(s) for figure 3:

**Figure supplement 1.** Comparing models of tACS modulation.

**Figure supplement 2.** Exploratory correlations between tACS modulation and FA values.

## Discussion

Using a combination of concurrent tACS-fMRI, cognitively precise behavioral outcomes, as well as effective and structural connectivity, we provide converging evidence for a causal role of aPFC-SMC connections in guiding action selection during emotional-action control (*Koch et al., 2018*). Emotional-action control improves when dual-site phase-coupled tACS is tuned to increase the efficacy of anterior prefrontal inhibition over sensorimotor cortex. Alternative interpretations of the findings, focused on transcutaneous entrainment (*Asamoah et al., 2019*) or retinal stimulation (*Schutter, 2016*), are ruled out by the experimental design. Namely, the in-phase and anti-phase conditions used identical stimulation parameters apart from their phase difference, thereby creating the same peripheral effects.

We did not observe the anticipated anti-phase tACS induced increase of congruency effects caused by a desynchronization of aPFC and SMC (*Polanía et al., 2015*). However, there was increased activity in contralateral aPFC and PPC for anti-phase trials as compared to in-phase trials. Speculatively, this increased BOLD signal could reflect the engagement of the control network contralateral to the stimulated circuit, compensating disruptive effects of anti-phase stimulation (*Sack et al., 2005*). Future studies could focus on characterizing these potential compensatory effects. This study grounds a human gain-of-function intervention on the known hierarchical organization of the frontal cortex (*Voytek et al., 2015*) and on phase-amplitude coupling as a mechanism for directional inter-regional neuronal communication (*Voytek et al., 2015*; *Canolty and Knight, 2010*). These findings provide the first stepping stone toward the development of interventions aimed to help patients suffering from social-emotional disorders. For instance, individuals with social anxiety are often unable to overcome avoidance tendencies, hampering interventions aimed to extinguish fear through exposure (*Craske and Stein, 2016*). Synchronizing aPFC-SMC theta-gamma coupling might temporally alleviate this lack of control (*Voytek and Knight, 2015*), allowing patients to benefit from exposure treatment. However, thus far it is unclear whether the findings reported here translate beyond healthy and highly educated male samples to more heterogeneous (healthy and patient) populations. Although the cognitive challenge provided by the AA task is rather mild, as is evidenced by the relatively low error rate (max. 20%), congruency effects elicited by the AA task do predict real-life emotional action control (*Kaldewaij et al., 2019*). Future studies could explore whether effects of in-phase stimulation hold for more challenging emotional control situations (e.g. *Sack et al., 2005*), extend to more heterogeneous populations, and can generate long-lasting effects through exposure treatment. Generally, the findings of this study pave the way for implementing physiologically-grounded non-invasive interventions aimed at reducing neural noise in cerebral circuits where communication relies on long-range phase-amplitude coupling.

## Materials and methods

### Participants

Forty-four male students of the Radboud University Nijmegen participated in this experiment after giving informed consent for participation and publication of anonymized data. Two participants were excluded because they did not attend the whole experiment; one participant was excluded because he failed to comply with the task instructions, yielding a total pre-determined sample of 41 participants. This sample size was determined on a priori estimates of statistical power calculated in

Gpower 3.1 (*Faul et al., 2007*) according to an expected effect size of *Cohen's d* = 0.4, as reported in- or calculated from *Volman et al., 2011a*; *Alekseichuk et al., 2016*. The a-priori sample size calculation, and other experimental details were preregistered before onset of data analysis, and after data collection, at the Open Science Framework, (https://osf.io/m9bv7/?view_only=18ccdf99a8e84a-faaebee393debbabe3). The study was approved by the local ethics committee (CMO2014/288). Only male participants were recruited to avoid having to control for potential variability that could be introduced by sex-related differences in hormone levels known to be associated with this task (*Volman et al., 2011b*) (for a full discussion see *Bramson et al., 2020*). Participants reported no history of mental illness or use of psychoactive medication. All participants had normal- or corrected to normal vision and were screened for contra-indications for magnetic resonance imaging and transcranial alternating current stimulation. The mean age was 23.8 years, SD = 3.4, range 18–34.

## Procedure

Data were acquired on 3 different days. On the first day, we acquired a structural T1 scan, a diffusion weighted imaging (DWI) scan, and a magnetic spectroscopy scan (MRS; not reported here). During the 2nd and 3rd day, electrodes were placed on the scalp, covering SMC and aPFC (see below for details on the precise localization). Afterward, participants performed a 5 min practice task, before starting an approach-avoidance task (35 min on each day) while receiving tACS stimulation and concurrently being scanned with fMRI.

## Approach-avoidance (AA) task

Participants performed a social-emotional approach-avoidance task that has previously been shown to require control over prepotent habitual action-tendencies to approach happy- and avoid angry faces (*Phaf et al., 2014*; *Roelofs et al., 2009*). Overriding these action-tendencies requires a complex form of cognitive control that operates on the interaction between emotional percepts and the emotional valence of the required actions (*Volman et al., 2011a*; *Roelofs et al., 2009*; *Volman et al., 2013*), and depends on aPFC control over downstream regions (*Koch et al., 2018*; *Volman et al., 2011a*; *Bramson et al., 2020*), implemented by theta-gamma phase-amplitude coupling (*Bramson et al., 2018*). Participants responded through a joystick with one degree of freedom (along the participant's midsagittal plane), holding the joystick with their right hand on top of their abdomen, while laying the MR-scanner and seeing a visual projection screen through a mirror system (see below). Participants were instructed to pull the joystick toward themselves when they saw a happy face, and push it away from themselves when they saw an angry face. These were the instructions in the congruent condition. In the incongruent condition, the participants were asked to push the joystick away from themselves when they saw a happy face, and pull it toward themselves when they saw an angry face (*Figure 1*). Written instructions were presented on the screen for a minimum of 10 s prior to the start of each block of 12 trials. Combined with the inter-block interval of around 20 s this gave at least 30 s washout of the stimulation. Congruent and incongruent conditions alternated between blocks. Trials started with a fixation cross presented in the centre of the screen for 500 ms, followed by the presentation of a face for 100 ms. Participants were asked to respond as fast as possible, with a maximum response time of 2000 ms. Movements exceeding 30% of the potential movement range of the joystick were taken as valid responses. Online feedback ('you did not move the joystick far enough') was provided on screen if response time exceeded 2000 ms. Each participant performed 288 trials on each of the 2 testing days, yielding 576 trials in total, equally divided over in-phase, anti-phase, and sham stimulation – as well as over congruent and incongruent conditions.

## tACS stimulation parameters

Transcranial alternating current stimulation (2 mA peak-to-peak) was applied using two sets of center-ring electrodes (80 mm inner Ø; 100 mm outer Ø; centre electrode had a Ø of 10 mm) (*Saturnino et al., 2017*). Stimulation was applied on-line during task performance in blocks of approximately 60 s (i.e. the length of a stimulus block; 12 trials), and consisted of electrical fields that changed polarity between the inner- and outer rings in theta-band (6 Hz) frequency over the aPFC and gamma-band (75 Hz), tapered with a 6 Hz sine wave over SMC. Gamma-band power was phase locked to peaks (in-phase) or troughs (anti-phase) of the theta-band signal, *Figure 2B*. Sham

consisted of an initial stimulation of 10 s to mimic potential sensations related to the onset of stimulation, after which stimulation was terminated. Stimulation conditions were alternated between blocks in pseudo-random fashion with the constraint that stimulation condition could never be repeated for two consecutive blocks. The position of the electrodes on the skull was determined for each participant by using individual structural T1 scans to which we registered masks of the regions of interest (MNI [−28–32 64] for SMC [*Bramson et al., 2018*]; and MNI [26 54 0] for lateral frontal pole (FPl) [*Bramson et al., 2020*; *Neubert et al., 2014*]). Precise placement of the centre electrode was achieved using Localite TMS Navigator (https://www.localite.de/en/products/tms-navigator/; RRID:SCR_016126). Electrodes were attached using Ten20 paste (MedCat; https://medcat.ccvshop.nl/Ten20-Pasta-Topf-4-Oz,−3er-Set) and impedance was kept below 10 kOhm (mean = 3.6, SD = 2.7). Stimulation was applied using two Neuroconn DC-PLUS stimulators (https://www.neuro-caregroup.com/dc_stimulator_plus.html; RRID:SCR_015520) that were placed inside a magnetically shielded box in the MR room. This box contained home-made electronics and BrainAmp ExG MR amplifiers (www.brainproducts.com) to continuously monitor the tACS output of the stimulator and filter out the RF pulses of the MR system.

tACS conditions (in-phase; anti-phase; sham) were applied within the same session with stimulation condition alternating between stimulus blocks (each lasting approximately 60 s; 12 trials), interleaved with periods of no stimulation (instructions between blocks; at least 30 s). Whereas previous studies have shown long lasting aftereffects of tACS, ranging up to 70 min (*Kasten et al., 2016*), we considered it unlikely that these would systematically bias the current results based on two arguments. First, reports showing strong aftereffects often implemented relatively long periods of continuous stimulation (*Alekseichuk et al., 2016*; *Kasten et al., 2016*; *Ruhnau et al., 2016*) necessary for the quantification of neural/behavioral stimulation effects. Short stimulation protocols comparable to the current study do not report aftereffects (*Johnson et al., 2020*; *Strüber et al., 2015*). tACS aftereffects would, by definition, influence neural activity in the sham epochs systematically succeeding each stimulation epoch. Yet, we observe strong differences in BOLD signal when comparing stimulation and sham epochs under the prefrontal electrodes (*Figure 1C*). Second, tACS aftereffects are thought to depend on synaptic plasticity rather than entrainment (*Vossen et al., 2015*). Possible offline plasticity aftereffects would build up across several minutes, that is, across stimulation periods with different phase relations, thus orthogonally to our experimental phase-based manipulation (*Vossen et al., 2015*). Aftereffects of entrainment are unlikely because entrainment is thought to fade out after several cycles (*Johnson et al., 2020*; *Strüber et al., 2015*; *Vossen et al., 2015*; *Halbleib et al., 2012*). Post-hoc analysis indeed revealed that behavioral congruency effects in the sham condition do not differ for trials preceded by in-phase as compared to anti-phase stimulation; t(40) = 0.4, p=0.7, arguing against systematic offline influence of tACS on behavioral congruency.

## Materials and apparatus

Magnetic resonance images were acquired using a 3T MAGNETROM Prisma MR scanner (Siemens AG, Healthcare Sector, Erlangen, Germany) using a 64-channel head coil with a hole in the top through which the electrode wires were taken out of the scanner bore.

The field of view of the functional scans acquired in the MR-sessions was aligned to a built-in brain-atlas to ensure a consistent field of view across days. Approximately 1800 functional images were continuously acquired in each scanning day using a multi-band six sequence, 2× 2× 2 mm voxel size, TR/TE = 1000/34 ms, Flip angle = 60°, phase angle p>>A, including 10 volumes with reversed phase encoding (A >> P) to correct image distortions.

High-resolution anatomical images were acquired with a single-shot MPRAGE sequence with an acceleration factor of 2 (GRAPPA method), a TR of 2400 ms, TE 2.13 ms. Effective voxel size was $1 \times 1 \times 1$ mm with 176 sagittal slices, distance factor 50%, flip angle 8°, orientation A ≫ P, FoV 256 mm.

Diffusion-weighted images were acquired using echo-planar imaging with multiband acceleration factor of 3. We acquired 93 1.6 mm thick transversal slices with voxel size of $1.6 \times 1.6 \times 1.6$ mm, phase encoding direction A >> P, FoV 211 mm, TR = 3350, TE = 71.20. 256 isotropically distributed directions were acquired using a b-value of 2500 s/mm$^2$. We also acquired a volume without diffusion weighting with reverse-phase encoding (p>>A).

## Behavioral analyses

We compared differences in error rates between congruent and incongruent trials for the different stimulation conditions whilst controlling for dose dependent effects of tACS ('tACS-dose') (*Evans et al., 2020*; *Kasten et al., 2019*). tACS-dose was estimated by taking BOLD contrast between stimulation and sham conditions extracted from right aPFC (*Evans et al., 2020*). This three way interaction (emotional-control*phase-condition*tACS-dose) was assessed using congruency effects estimated from participant-by-participant averages (with RM-ANOVA), as well as trial-by-trial data (with Bayesian mixed effect models). RM-ANOVA was used to facilitate comparisons with earlier studies reporting on this task (e.g. *Bramson et al., 2018*; *Volman et al., 2011a*; *Volman et al., 2013*) and implemented in JASP (https://jasp-stats.org/; RRID:SCR_015823). We used Bayesian mixed effect models (implemented in *R* 3.5.3 using the *brms* package [*Bürkner, 2017*]) because they are robust to potential violations of normality or homoscedasticity. The Bayesian mixed models included random intercepts for all subjects and random slopes for all fixed effects (congruency- and stimulation condition) and their interaction per participant. This model adheres to the maximal random effects structure (*Barr, 2013*). Outputs of these models are log odds with credible intervals ('*B*'). In these analyses, an effect is seen as statistically significant if the credible interval does not contain zero with 95% certainty.

We hypothesized that the congruency effect in error rates would decrease for in-phase condition and increase for anti-phase stimulation and that the size of the effect per participant would depend on the BOLD effect of tACS versus sham, a measure of dose dependence that is orthogonal to the contrast of interest (in-phase versus anti-phase). These expectations were preregistered at the Open Science Framework: (https://osf.io/m9bv7/?view_only=18ccdf99a8e84afaaebee393debbabe3).

## Modeling of stimulation currents

Current density under the electrodes was simulated using SIMNIBS version 3.1 (https://simnibs.github.io/simnibs/build/html/index.html; *Thielscher et al., 2015*). We used the template head model and standard conductivities provided by SIMNIBS. Electrodes were placed over SMC and aPFC and direct currents of 1 mA were estimated to run from the inner electrode toward the outer ring. Electrode placement and current density estimates for both electrode pairs are visualized in *Figure 2A*.

## fMRI analyses – data preprocessing

All processing of the images was performed using MELODIC 3.00 as implemented in FSL 6.0.0 (https://fsl.fmrib.ox.ac.uk). Images were motion corrected using MCFLIRT (*Jenkinson et al., 2002*), and distortions in the magnetic field were corrected using TOPUP (*Andersson et al., 2003*). Functional images were rigid-body registered to the brain extracted structural image using FLIRT. Registration to MNI 2 mm standard space was done using the nonlinear registration tool FNIRT. Images were spatially smoothed using a Gaussian 5 mm kernel and high pass filtered with a cut-off that was automatically estimated based on the task structure. Independent component analysis was run with a pre-specified maximum of 100 components (*Beckmann and Smith, 2004*); these components were manually inspected to remove potential sources of noise; (*Griffanti et al., 2017*). fMRI analyses – signal-to-noise (SNR) artefacts resulting from electrode presence.

To assess whether the presence of electrodes on the scalp had an effect on the signal-to-noise ratio in the fMRI signal, we estimated temporal signal-to-noise ratio (tSNR) from the right aPFC (electrode present) to the left aPFC (electrode not present). tSNR was calculated by dividing the mean of the signal over time by the standard deviation, separately for the left and right aPFC. These estimates were extracted from a mask of the lateral frontal pole (*Neubert et al., 2014*). We also compared the mean signal extracted from the right to the left aPFC.

## fMRI analyses - GLM

First and second level GLM analyses were performed using FEAT 6.00 implemented in FSL 6.0.0. The first-level model consisted of 12 task regressors: Approach angry, approach happy, avoid angry, and avoid happy trials were modelled separately for each stimulation condition (in-phase, anti-phase and sham). In each regressor, each event covered the time interval from presentation of a face until the corresponding onset of the joystick movement. Estimated head translations/rotations during scanning (six regressors), temporal derivatives of those translations/rotations (six regressors), and

MR-signals in white matter and cerebrospinal fluid (two regressors) were included to the GLM as nuisance covariates. We considered the following comparisons. Emotional control effects were estimated by comparing incongruent trials (approach angry and avoid happy) to congruent trials across the three stimulation conditions (total congruency effect), as well as separately for each stimulation condition (in-phase, out-of-phase, and sham congruency effects). Overall stimulation effects were estimated by comparing in-phase and anti-phase stimulation conditions to the sham condition, aggregated across congruent and incongruent conditions. We reasoned that this contrast would combine potential entrainment and compensatory mechanisms and thereby result in an unbiased estimate reflecting the extent to which the underlying neural tissue was perturbed by the stimulation; yielding a 'tACS-dose' measure. Phase-dependent stimulation effects were estimated by comparing in-phase stimulation to anti-phase stimulation, across congruent and incongruent conditions. First-level models of the two separate sessions were combined using fixed effects analyses implemented in FEAT. Group effects were assessed using FLAME one with outlier de-weighting (*Woolrich et al., 2004*), making family-wise error corrected cluster-level inferences using a cluster-forming threshold of z > 2.3. This threshold provides a false error rate of around 5% when using FSL's FLAME 1 (*Eklund et al., 2015*).

## fMRI analyses – dynamic causal modeling

Dynamic causal modeling (DCM 12.5), implemented in SPM 12, was used to make inferences on network effects of the tACS manipulation (*Friston et al., 2003*). Dynamic causal modeling is an approach that aims to infer hidden neuronal dynamics from neuroimaging data (*Friston et al., 2003*; *Stephan et al., 2010*). It requires the specification of plausible generative models describing how neural activity leads to observed neuroimaging data through hemodynamic functions. This approach allows formal comparison of different models explaining the same data (*Stephan et al., 2009*; *Penny et al., 2010*). DCM provides estimates of effective connectivity between neural populations, and its modulation by experimental conditions (e.g. in this case tACS present or absent). We preregistered the hypothesis that changes in connectivity between aPFC and SMC due to tACS might be mediated by the amygdala, a region linked to the regulation of social-emotional action tendencies (*Volman et al., 2011a*; *Kaldewaij et al., 2019*; *Bramson et al., 2020*). To test this hypothesis, we constructed a model space in which each model contained three regions; the left amygdala (see GLM results); the right aPFC; and the left SMC.

The regions of interest were defined separately for each participant based on their functional effects (from the GLM analyses) but constrained to a-priori determined regions; right aPFC, based on the FPl mask (*den Uyl et al., 2015*; *Voytek et al., 2015*); left SMC, based on a previous study involving the same task [MNI: −28–32 64] (*Bramson et al., 2018*); and the left amygdala, based on the automated anatomical labeling mask (*Volman et al., 2011a*; *Kaldewaij et al., 2019*; *Tzourio-Mazoyer et al., 2002*). The timeseries of the first eigenvariate across all significant voxels in the ROI for the amygdala and SMC were extracted from a sphere with 3 mm radius (amygdala) and 5 mm radius (SMC) around the peak voxel detected in each ROI in a GLM F-contrast across effects of interest. aPFC time series were extracted from a sphere with 3 mm radius around the peak voxel in the incongruent >congruent t-contrast in the right lateral frontal pole (*Neubert et al., 2014*).

Model structure for all models under comparison was based on the minimal architecture needed to dissociate whether synchronization between aPFC and SMC was gated by the amygdala or via other region(s). All models consisted of a directed connection between the amygdala and SMC (*Grèzes et al., 2014*) (indicated as amygdala→SMC, amygdala → aPFC, and aPFC → amygdala [*Bramson et al., 2020*; *Folloni et al., 2019*]). A subset of models also included aPFC → SMC (*Figure 3*), which captures the connectivity between aPFC and SMC gated via cerebral structures other than the amygdala given that the two structures do not share a monosynaptic connection (*Neubert et al., 2014*). Return connections SMC → amygdala and SMC → aPFC were not included. We assumed these to be unnecessary to explain the effects of synchronization given the hierarchical nature of prefrontal control (*Badre and D'Esposito, 2009*; *Badre and Nee, 2018*). The onset of the stimulus in each trial (presentation of the face) was taken as a driving input (DCM.C matrix) feeding into the amygdala node of each model. Emotional control (all incongruent trials) and tACS condition (all in-phase trials, all anti-phase trials, or a combination; *Figure 3—figure supplement 1*) were allowed to modulate different connections (DCM.B matrix).

We created 84 models that systematically varied in four dimensions of interest, which were compared using family-wise model comparison (*Penny et al., 2010*). Dimension 1 was the modulation of emotional control, where control was allowed to modulate either amygdala → aPFC connection or aPFC self-connections (*Volman et al., 2013*). Dimension 2 was the connectivity structure of the models. This dimension consisted of two different model types: with or without a direct connection from aPFC → SMC (*Figure 3—figure supplement 1A*). The presence of this connection can account for influence of aPFC on SMC that is not gated by the amygdala. Dimension 3 reflects the nature of the tACS effects (*Figure 3—figure supplement 1B*; *Piray et al., 2017*). We modeled no effect of tACS (Stim 0); only the in-phase condition (Stim 1); only the anti-phase condition (Stim 2); both conditions separately (Stim 3); both conditions with opposite sign and similar amplitude (Stim 4). Dimension 4 arbitrated over the location of tACS modulation (*Figure 3A*, *Figure 3—figure supplement 1C*), which was allowed to modulate the amygdala → SMC, aPFC → amygdala, and aPFC → SMC, or any combination of the three connections. We differentiated these possibilities by separating all models into model families for each dimension, allowing inferences per dimension while averaging over all other parameters (*Figure 3—figure supplement 1*; *Penny et al., 2010*).

After model comparison, we extracted connectivity estimates that were altered by tACS from the winning model and used those parameters to predict tACS effects on behaviour. The estimated changes in connectivity due to tACS are estimated independently of the behavioral effects of tACS, and were used to test for an interaction between emotional control (congruent, incongruent), stimulations phase (in-phase, anti-phase) and effective connectivity (DCM.B matrix) using RM-ANOVA and Bayesian mixed effects models.

## DWI analyses

All analyses of diffusion data were performed in FSL's FDT 3.0 (https://fsl.fmrib.ox.ac.uk). We used TOPUP to estimate susceptibility artifacts using additional b = 0 volumes with reverse-phase coding direction (*Andersson et al., 2003*). Next, EDDY correction was used (using the fieldmap estimated by TOPUP) to correct distortions caused by eddy currents and movement (*Andersson and Sotiropoulos, 2016*). We used BedpostX to fit a crossing fiber model with three fiber directions (*Behrens et al., 2007*). Connections between aPFC and BA6 were reconstructed using *probtrackx2*. Seed mask was the lateral frontal pole (*Neubert et al., 2014*), which is a mask that contains voxels bordering white matter. Target mask was area BA6 contained within the Juelich atlas in FSL. This probabilistic mask was thresholded to contain only voxels that were at least 50% likely to be in BA6. Tractography was run twice with the standard settings recommended in probtrackX and an exclusion mask in the midline. Estimated connection strengths were normalized to unit length within participants. We averaged all connections over participants and thresholded this volume at 5 to create a mask of likely pathways linking FPl (within aPFC) and BA6. In the second step, we constrained the tractography to either only include connections via the thalamus, or only via medial and lateral pathways through the prefrontal cortex.

Tract-based spatial statistics (TBSS; *Smith et al., 2006*) was used to assess whether the white matter integrity in the voxels in this connection mask explained part of the variance in response to tACS. *TBSS* was run using the default settings provided by FSL. In short, FA images were eroded and nonlinearly registered to the standard space. Afterward, we derived a mean skeleton based on all participants and thresholded the skeletonized FA values at 2. We then used *randomize* (*Winkler et al., 2014*) with threshold-free cluster enhancement (*Smith and Nichols, 2009*) to make inferences on the correlation between FA integrity and DCM effective connectivity estimates (in-phase versus anti-phase). To reduce the search space, the comparison was constrained to voxels that were part of the mask linking FPl to BA6 with and without the constrained connectivity (see *Figure 3—figure supplement 2*).

## Acknowledgements

The authors would like to thank Freek Nieuwhof and Uriel Plones for their help in setting up the tACS-fMRI setup and Amy Abelmann for her assistance in data collection. This work was supported by a VICI grant (#453-12-001) from the Netherlands Organization for Scientific Research (NWO) and a consolidator grant from the European Research Council (ERC_CoG-2017_772337) awarded to Karin Roelofs.

## Additional information

### Funding

| Funder | Grant reference number | Author |
| --- | --- | --- |
| Nederlandse Organisatie voor Wetenschappelijk Onderzoek | (#453-12-001 | Karin Roelofs |
| European Commission | ERC_CoG-2017_772337 | Karin Roelofs |

The funders had no role in study design, data collection and interpretation, or the decision to submit the work for publication.

### Author contributions

Bob Bramson, Conceptualization, Data curation, Formal analysis, Investigation, Visualization, Methodology, Writing - original draft, Project administration, Writing - review and editing; Hanneke EM den Ouden, Methodology, Writing - review and editing; Ivan Toni, Conceptualization, Formal analysis, Supervision, Visualization, Methodology, Writing - original draft, Writing - review and editing; Karin Roelofs, Conceptualization, Formal analysis, Supervision, Funding acquisition, Visualization, Methodology, Writing - original draft, Project administration, Writing - review and editing

### Author ORCIDs

Bob Bramson https://orcid.org/0000-0002-0752-0805
Hanneke EM den Ouden http://orcid.org/0000-0001-7039-5130
Ivan Toni https://orcid.org/0000-0003-0936-3601

### Ethics

Human subjects: The study was approved by the local ethics committee (CMO2014/288). All participants gave informed consent for participation and publication of anonymized data.

### Decision letter and Author response

Decision letter https://doi.org/10.7554/eLife.59600.sa1
Author response https://doi.org/10.7554/eLife.59600.sa2

## Additional files

### Supplementary files

• Transparent reporting form

### Data availability

Data used for all reported analyses are available from the donders data repository: data.donders.ru.nl, collection identifier: di.dccn.DSC_3023010.01_369.

The following dataset was generated:

| Author(s) | Year | Dataset title | Dataset URL | Database and Identifier |
| --- | --- | --- | --- | --- |
| Bramson B, Toni I, Roelofs K | 2020 | Influencing control over social-emotional actions using tACS-fMRI | https://data.donders.ru.nl/collections/di/dccn/DSC_3023010.01_369 | Donders Repository, DSC_3023010.01_369 |

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

# Appendix 1

## Supplementary Results

Participants performed a social-emotional approach-avoidance task in which they approached or avoided happy and angry faces using a joystick. Approaching angry- and avoiding happy faces is incongruent with automatic action tendencies to approach appetitive and avoid threatening stimuli, and requires control over these prepotent habitual action tendencies (*Figure 1*; *Volman et al., 2011a*; *Bramson et al., 2020*; *Roelofs et al., 2009*). fMRI was acquired during task performance whilst participants received on-line dual-site theta-gamma tACS over aPFC and SMC, *Figure 2A;B*. Control over social-emotional action tendencies involves phase-amplitude coupling between prefrontal theta- and SMC gamma-band rhythmic activity (*Bramson et al., 2018*), the phase organisation of which is best reflected by the in-phase condition. In-phase, anti-phase and sham conditions (*Figure 2B*), were alternated throughout the task, allowing comparison of on-line, within-subject effects of stimulation conditions.

### Signal quality below the electrodes

There was a reduction in absolute signal in right aPFC below the frontal electrode as compared to the contralateral aPFC, t(40) = 5.5, p<0.001. This consisted of an average reduction in signal intensity of 11% (SD 10%). However, there was no difference in temporal SNR in right aPFC as compared to left; t(40) = 0.2, p=0.7. These results are in line with earlier reports on noise induced by the presence of electrodes and stimulation equipment in the scanner bore (*Antal et al., 2014*). Although the absolute MR signal is reduced due to the presence of the tACS electrode, there is no effect on the SNR of the underlying brain region, suggesting that condition differences can be estimated reliably. This is supported by the consistency between the results obtained in the congruency contrast across stimulation conditions, and the results obtained in earlier reports using the same paradigm (*Figure 1C*, *Figure 2—figure supplement 1*; *Volman et al., 2011a*; *Bramson et al., 2020*; *Roelofs et al., 2009*)

### Behavioral and neural costs of controlling social-emotional action tendencies

Across all conditions (stimulation and sham combined), reaction times were longer in the incongruent (M = 687 ms, SD = 141) than in the congruent condition (M = 637 ms, SD = 111), *brms* estimate B = 23.1 ms, credible interval (*CI*) [15 31]; paired t-test congruent versus incongruent t(40) = 5.9, p<0.001, Cohen's d (*M1-M2/SD$_{pooled}$*) = 0.39 [.22 1]. Participants also made more errors in the incongruent condition (94.8% correct, SD = 3.4%) than in the congruent (96.9% correct, SD = 2.3%), this was significant for both aggregated scores: t(40) = 5.4, p<0.001, d = 0.7 [.01 1.35], as well as on the trial-level: *B = 0.29 CI [.16. 42]*. The latter parameter is a log odds and indicates that participants are more likely to make correct responses in the congruent than incongruent trials. These effects illustrate the behavioral cost of controlling automatic action tendencies.

Whole-brain cluster-corrected fMRI analyses show that controlling social-emotional action tendencies elicited stronger BOLD activity in bilateral prefrontal cortex: MNI [−18 54–16] and [40 42 32], extending into lateral frontal pole (*Bramson et al., 2020*; *Neubert et al., 2014*); bilateral posterior parietal cortex (PPC) MNI: [38 -58 38] and [−56–44 46]; precuneus [6 -62 72], temporal cortex [−66–48 0]; and inferior frontal gyrus [60 20 4], *Figure 1C*, *Figure 2—figure supplement 1C*. Activity in the parahippocampal gyrus [−24–40 −14], extending into hippocampus and amygdala, medial prefrontal cortex [0 66 2], and lateral occipital cortex [52 -74 -10] was stronger for congruent than for incongruent trials. Similar effects have been reported in several earlier reports (*Volman et al., 2011a*; *Kaldewaij et al., 2019*; *Bramson et al., 2020*), with prefrontal effects depending crucially on the interaction between the emotional content of the action and the emotional content of the percept, over and above the emotional value of the action or stimulus alone (*Volman et al., 2011a*; *Volman et al., 2011b*; *Roelofs et al., 2009*).

## Cerebral effects of dual-site phase-coupled tACS versus Sham

During task performance each participant received concurrent transcranial alternating current stimulation on two locations; aPFC (*Bramson et al., 2020*; *Neubert et al., 2014*) and sensorimotor cortex, *Figure 2A*;B. These locations, frequency, and phase relationship were based directly on a previous study showing aPFC-SMC phase-amplitude coupling in theta-gamma band rhythmic signals (*Bramson et al., 2018*). Whole-brain cluster-corrected analyses contrasting both stimulation conditions versus sham showed increased activity in several brain regions, most of which overlapped with task relevant regions: right prefrontal cortex [42 16 48], extending into FPl [22 60 0]; bilateral posterior parietal cortex [50 -58 50] and [−56–44 46], *Figure 2C*; *Figure 2—figure supplement 1C*;D. These effects support previous experimental and theoretical work suggesting that tACS effects ongoing neural activity and effects are task-dependent (*Ruhnau et al., 2016*).

## Cerebral effects of in-phase versus anti-phase dual-site phase-coupled tACS

Contrasting in-phase with anti-phase conditions showed stronger activity in left aPFC [−28 50–10], contralateral to stimulated right aPFC, and left PPC [−56–46 52], *Figure 2—figure supplement 1E*. These results suggest that desynchronising right aPFC and left SMC might induce compensatory effects in the contralateral homotopic control regions. These findings illustrate the importance of measuring online effects of brain stimulation because stimulation might have distal effects on neural tissue (*de Graaf and Sack, 2018*).

## Dose dependent effects of tACS on social-emotional control

To assess whether the tACS intervention increases control over social-emotional behaviour, we compared behavioral congruency effects between different stimulation conditions whilst controlling for dose dependent effects. Dose dependent responses were calculated by extracting the differential BOLD signal under the prefrontal stimulated region (right aPFC, [22 60 0]) when comparing stimulation versus sham epochs. This BOLD effect provides a participant-by-participant index of aPFC response to the tACS intervention that is orthogonal to the in-phase versus anti-phase comparison.

Following the expectation preregistered at the Open Science Framework: (https://osf.io/m9bv7/?view_only=18ccdf99a8e84afaaebee393debbabe3), there was a three way interaction between emotional control, phase condition, and tACS-dose, both on the trial level; $B = 0.1\ CI\ [.007.\ 2]$, and in aggregated error rates; RM-ANOVA, $F(1,39) = 9.3$, p=0.004, partial eta$^2$ = 0.19, indicating that participants performed differently in the two stimulation conditions as a function of the strength of the tACS influence. Post-hoc correlation analyses between BOLD change due to tACS and congruency effects showed that these effects are mainly driven by the in-phase condition: Spearman's Rho = 0.32, p=0.04, *Figure 2B*. This finding indicates that, in the in-phase condition, larger BOLD responses to the inhibitory theta-band rhythm corresponded to smaller behavioral congruency effects. In the anti-phase condition, the correlation between tACS-dose and behavioral congruency effects was not significant (Spearman's Rho = −0.23, p=0.14). However, the sign of the correlation matches the preregistered expectations: increased entrainment during anti-phase stimulation (reflected in lower BOLD response during theta-band stimulation; *Behrens et al., 2007*) would lead to larger congruency effects due to desynchronization between aPFC and SMC. Differences in BOLD signal between anti-phase and in-phase conditions (*Figure 2—figure supplement 1E*) could reflect the engagement of the control network contralateral to the stimulated circuit. This engagement might reflect compensatory mechanisms counteracting disruptive effects of anti-phase stimulation (*Sack et al., 2005*; *de Graaf and Sack, 2018*), thereby obscuring the relationship between tACS-dose and increases in behavioral congruency. However, the BOLD signal difference between anti-phase and in-phase conditions extracted from aPFC and PPC (*Figure 2—figure supplement 1E*) did not correlate with anti-phase behavioral congruency effects (both r(39) <.14, p>0.3). Controlling for baseline congruency effects (in the sham condition) did not change these results; Rho = 0.32, p=0.044 for in-phase; Rho = −0.26, p=0.09 for anti-phase. These results suggest that theta-gamma phase-coupled tACS increases control over social-emotional behaviour in participants that respond to stimulation. Contrasting error-rate congruency effects (% correct for congruent - % correct for incongruent trials) without accounting for the dose dependent responses did not show a statistically reliable group effect of stimulation condition on the trial level, $B = 0.05\ CI\ [−0.05.\ 14]$, nor on the

aggregated scores RM-ANOVA; *F(1,39)* = 3.75, p=0.06, partial eta$^2$ = 0.088, *Figure 2—figure supplement 1A*.

Reaction times did not show differential effects in the different stimulation conditions, emotional control*phase condition*tACS-dose; *F*(1,40) = 0.05, p=0.8, congruency*stimulation; *F*(1,41) = 2.3, p=0.14, *Figure 2—figure supplement 1B*. The lack of effects on reaction times is in line with expectations preregistered at the Open Science Framework: (https://osf.io/m9bv7/?view_only=18ccdf99a8e84afaaebee393debbabe3) as well as earlier stimulation studies using this same task (*Volman et al., 2011a*). This finding suggests that the aPFC control required during the AA task operates at the level of action selection.

## tACS changes aPFC-SMC effective connectivity, which correlates with behavioral effects

To assess the network effects of dual-site tACS, we used dynamic causal modeling (DCM; *Friston et al., 2003*) verifying whether aPFC→SMC connectivity was gated by the amygdala. We created a set of 84 models with two different connectivity structures; with and without direct connection between aPFC and SMC, *Figure 3—figure supplement 1A*. These models differed in the way tACS could affect connectivity, *Figure 3—figure supplement 1B*. Possibilities consisted of no effect of stimulation (Stim 0), effects of one stimulation condition (Stim 1 and Stim 2, Sup *Figure 2B*), independent effects of both conditions (Stim 3) or opposite sign but equal amplitude effects of both stimulation conditions (Stim 4). Finally, models varied with respect to the location of the tACS modulation: amygdala → SMC, aPFC → amygdala or aPFC → SMC, two of those three connections, or all three connections, *Figure 3—figure supplement 1C*.

Family comparisons showed that the structural model containing a direct connection from aPFC→SMC was more likely than a model without this connection, *Figure 3—figure supplement 1A*. This suggests that not all effects of aPFC→ SMC are mediated by the amygdala. Comparing the different possible effects of stimulation showed that models in which both stimulation conditions were modeled separately had the strongest model evidence, *Figure 2B*. Finally, comparisons over possible locations where tACS could modulate connectivity showed tACS effects on all three modeled connections, *Figure 3—figure supplement 1C*

We extracted connectivity parameters from all three modulated connections and used them to account for differences in behavioral response. This showed that differences in effective connectivity between in- and anti-phase on the aPFC → SMC connection predicted differences in behavioral effects between stimulation conditions both on the trial level; *B = 0.11 CI [.02. 2]* and on aggregated congruency-effects; RM-ANOVA; *F(1,39)* = 6.01 *p* = 0.019, np$^2$ = 0.13. Over the whole group connectivity parameters did not differ between in- and anti-phase conditions, *t(40),=1.1, p=0.27*. Post-hoc tests showed that those participants that showed more negative connectivity between aPFC→SMC in the in-phase as compared to anti-phase condition showed a reduction in congruency effects in the in-phase as compared to the anti-phase condition, t(21) = −2.9, p=0.008. This was not the case for participants with no difference, or more negative connectivity for anti-phase versus in-phase, p=0.35; main *Figure 3B*. This improvement of control in the in-phase condition remains when outliers in behavioral congruency are removed, *t(20),=−2.7, p=0.012*, and if we remove the participants that did not show a response to amygdala input in the DCM analysis (n = 3), *t(19) = −2.8, p=0.01*. Strikingly, for the participants with stronger negative connectivity between aPFC→SMC the congruency effect was no longer significantly different from zero, *t(21),=0.99 p=0.33* in the in-phase condition, indicating that they did no longer perform worse when having to override automatic action-tendencies. Participants that did not show tACS modulation of aPFC→SMC connections in the in-phase condition did show a congruency effect *t(18)* = 2.6, p=0.018. These effects suggest that successful synchronization of aPFC-SMC rhythmic activity increases control over social-emotional behavior by increasing aPFC inhibition over SMC.

To explore whether structural characteristics might underlie individual differences in response to stimulation, we tested whether differences between aPFC→SMC connectivity parameters between in-phase and anti-phase conditions correlated with FA integrity in voxels potentially connecting aPFC to SMC. We observed a correlation between FA integrity and DCM connectivity estimates for in-phase versus anti-phase in white matter regions leading up to right BA6, *tfce* corrected for voxels in a thresholded mask connecting FPl (aPFC) and BA6; p=0.046, *Figure 3—figure supplement 2*.

Here increased inhibition in the in-phase condition (relative to anti-phase) is related to higher FA integrity in white matter leading up to BA6. Further exploration showed that this correlation is present when considering cortico-thalamo-cortical connections; corrected p<0.048 but not when only considering cortico-cortical connections. These findings suggest that tACS synchronization of aPFC→SMC connectivity might partly depend on the structural integrity in aPFC-thalamic-SMC connectivity (*Halassa and Kastner, 2017*). However, given that these results are exploratory, seem to be driven by a relatively small number of participants (*Figure 3—figure supplement 2B*), and depend on the threshold chosen for creating the masks, they need to be considered with caution and warrant replication.

