## [Decision Letter]

**Acceptance summary:**

Bramson et al. investigate the neural basis of emotional-action control. Building on their prior work, the authors hypothesized that coupling between the anterior prefrontal cortex (aPFC) and sensorimotor cortex (SMC) subserves emotion-action control. Here, they set to manipulate this hypothesized mechanism using dual-site transcranial alternating current stimulation (tACS). Results reveal a dose-dependent effect of in-phase tACS on emotional action control, suggesting that aPFC-SMC coupling does indeed play a mechanistically crucial role in emotional action selection.

**Decision letter after peer review:**

Thank you for submitting your article "Improving emotional-action control by targeting long-range phase-amplitude neuronal coupling" for consideration by *eLife*. Your article has been reviewed by two peer reviewers, and the evaluation has been overseen by a Reviewing Editor and Richard Ivry as the Senior Editor. The reviewers have opted to remain anonymous.

The reviewers have discussed the reviews with one another and the Reviewing Editor has drafted this decision to help you prepare a revised submission.

Summary:

Building on recent work by Voytek and others, Bramson et al. investigate the neural basis of emotional-action control continuing their previous line of research. In prior work (Bramson et al., 2018), the authors hypothesized that theta-gamma coupling between the anterior prefrontal cortex (aPFC) and sensorimotor cortex (SMC) subserves emotion-action control. Here, they set out to manipulate this mechanism to show its causal involvement. The authors use the approach-avoidance task separating emotionally-congruent and incongruent action tendencies while applying dual-site transcranial alternating current stimulation (tACS) and monitoring neural activity using fMRI. Importantly, stimulation is set to either facilitate aPFC-SMC theta-gamma phase-amplitude coupling (in-phase) or disrupt it (anti-phase). Results reveal a dose-dependent effect of in-phase tACS on action control, suggesting that aPFC-SMC coupling does indeed play a critical role in action selection.

The reviewers and I were enthusiastic about the report.…

• The study is compelling and innovative. The authors use a multimethod approach combining brain stimulation, functional neuroimaging, diffusion-weighted imaging, and dynamic causal modeling to assemble an in-depth picture of emotional-action control mechanism in the human brain.

• The findings lay out the path for the potential application of tACS in anxiety therapy.

• I find the study interesting and contributing to scientific knowledge

• This report is an elegant and novel demonstration of a gain-of-function manipulation that improves control over emotional action tendencies via the delivery of alternating transcranial current to the cortical elements of a circuit hypothesized to play a role in emotional regulation.

• Particular strengths include an intuitive and established behavioral assay, multiple within-subject controls for stimulation parameters, and the simultaneous use of multiple techniques allowing the authors to establish a 'dose-dependent' effect of aPFC inhibition.

• Overall, the data presented here are a strong addition to the literature

Essential revisions:

Despite their enthusiasm, we do have a few questions about the interpretation, and some suggestions for further strengthening the report.

1) Approach. The decision to intermix stimulation protocols within a session needs to be better motivated.

• The authors use a within-subjects design without separating different stimulation protocols. The three stimulation conditions – in-phase, anti-phase, and sham tACS – are applied within the same session, in a block-design manner. Typically, within-subjects studies separate stimulation sessions by 48 hours or more to avoid carryover effects (Heise et al., 2019; Mansouri et al., 2019; Amador de Lara et al., 2018; Alekseichuk et al., 2016).

• It has been shown that tACS can induce aftereffects (for review, Vosskuhl et al., 2018).

• If here the authors made a design decision to mix together different stimulation conditions within a session, this decision needs to be justified. Providing the rationale behind the experimental design would benefit the paper and increase the credibility of the findings.

2) Approach/Results. Authors should provide evidence that the key results are not contaminated by potential carry-over effects from preceding blocks

3) Results. Figure 2D raises the possibility that the anti-phase condition produces a correlation between aPFC BOLD and the congruency effect that is roughly equal and opposite of what is observed in the in-phase condition (as opposed to producing no correlation, for instance). If this is a reasonable interpretation, then this potentially intriguing effect should be noted in the Results and noted in the Discussion.

4) Results. In general, the paper should be revised to clearly indicate the polarities and valence of tACS stimulation, BOLD effects, and changes in congruency effect.

• What is the difference between negative tACS dose in-phase and positive tACS dose anti-phase stimulation? The scatter plot and regression line in Figure 2D do not clearly communicate the difference between in-phase and anti-phase tACS with complementary tACS dose.

5) Discussion. Null results, and potential implications for major take-home points. Authors need to provide some discussion of their unsupported predictions.

• In their preregistered hypothesis (Open Science Framework), the authors predict an increase of congruency effect in the anti-phase stimulation condition (subsection “Behavioural analyses”, last paragraph), however, this expectation was not confirmed by the results (subsection “Dose dependent effects of tACS on social-emotional control”, second paragraph). The absence of the effect of anti-phase tACS that was expected to be in the opposite direction to the effect of in-phase stimulation raises questions about the causal involvement of aPFC-SMC coupling in emotional-action control.

• It is clear from the results that strengthening aPFC-SMC coupling enhances control.

• However, if the disruption of such in-phase coupling by the anti-phase stimulation doesn't lead to the reduction in control it suggests that some other mechanisms might play a compensatory role. I recommend addressing this issue in the discussion of the results.

6) Discussion/Abstract. The authors emphasize the clinical implications, but this discussion should be qualified and softened.

• In this experiment, a substantial change in brain activity (as measured by BOLD) leads to a reliable, but relatively subtle behavioral change in a task that asks participants to override a mild response tendency. Indeed, according to Figure 1B, even subjects performing relatively poorly in the incongruent condition are still getting it right more than 80% of the time, suggesting that this condition presents no more than a mild challenge.

• Emotional control will inevitably be harder in a real world scenario in which individuals will certainly perceive there to be far more at stake – this will be doubly true for any clinical population.

• Thus, claims like "Similar interventions could help patients suffering from emotional-social disorders" and others should be softened. The authors present fascinating research that should be characterized more as a first step in developing gain-of-function therapeutics, instead of a treatment ready for, or even on the cusp of, application in the clinic.

---

## [Author Response]

Essential revisions:Despite their enthusiasm, we do have a few questions about the interpretation, and some suggestions for further strengthening the report.1) Approach. The decision to intermix stimulation protocols within a session needs to be better motivated.• The authors use a within-subjects design without separating different stimulation protocols. The three stimulation conditions – in-phase, anti-phase, and sham tACS – are applied within the same session, in a block-design manner. Typically, within-subjects studies separate stimulation sessions by 48 hours or more to avoid carryover effects (Heise et al., 2019; Mansouri et al., 2019; Amador de Lara et al., 2018; Alekseichuk et al., 2016).• It has been shown that tACS can induce aftereffects (for review, Vosskuhl et al., 2018).• If here the authors made a design decision to mix together different stimulation conditions within a session, this decision needs to be justified. Providing the rationale behind the experimental design would benefit the paper and increase the credibility of the findings.

We agree that we could better clarify the design decision to mix different stimulation conditions within a session, and why tACS aftereffects are unlikely to explain the findings. To reiterate the context, the study is focused on measuring behavioral and fMRI effects *during* the delivery of tACS. Since tACS can induce aftereffects 1-70 minutes post stimulation (e.g. Strüber et al., 2015; Vossen et al., 2015; Kasten et al., 2016; Vosskuhl et al., 2018), it is relevant to discuss whether those aftereffects contribute to the reported findings.

tACS aftereffects are thought to depend mostly on synaptic plasticity (Vossen et al., 2015), and they are not observed for very short periods of stimulation as used in the current study (Strüber et al., 2015). The current study combines short periods of stimulation (1 min) with washout periods of at least 30 seconds between successive stimulation blocks. Studies that report aftereffects of tACS (e.g. Neuling et al., 2013; Alekseichuk et al., 2016; Kasten et al., 2016) use relatively long periods of continuous stimulation (at least > 10 minutes, reviewed in Strüber et al., 2015). The empirical evidence collected during this experiment confirms that plasticity induced aftereffects, if present, are marginal. Namely, tACS aftereffects would influence neural activity in the sham epochs systematically succeeding each stimulation epoch. Yet, we observe strong differences in BOLD signal when comparing stimulation and sham epochs under the prefrontal electrodes (Figure 1C).

Importantly, the experiment was designed to avoid systematic aftereffect biases in the main finding of this study. The different conditions (in-phase, anti-phase, and sham) were rapidly alternated. Plasticity aftereffects would require several minutes to build up (Vossen et al., 2015), and that implies that potential aftereffects would stretch over both in-phase and anti-phase conditions. New analyses show that behavioral congruency effects in the sham condition do not differ for trials preceded by in-phase as compared to anti-phase stimulation; t(40) = 0.4, p = 0.7, arguing against systematic offline influence of tACS on behavioral congruency. Aftereffects due to rhythmic entrainment are also unlikely. Entrainment is thought to fade out after a few cycles following the entraining stimulation (Halbleib et al., 2012; Strüber et al., 2015; Vossen et al., 2015), i.e. over a time-frame much shorter than the wash-out epochs interposed between the different stimulation conditions.

We have incorporated these considerations in the manuscript:

“tACS conditions (in-phase; anti-phase; sham) were applied within the same session with stimulation condition alternating between stimulus blocks (each lasting approximately 60 seconds; 12 trials), interleaved with periods of no stimulation (instructions between blocks; at least 30 seconds). Whereas previous studies have shown long lasting aftereffects of tACS, ranging up to 70 minutes (Kasten et al., 2016), we considered it unlikely that these would systematically bias the current results based on two arguments. […] Post-hoc analysis indeed revealed that behavioral congruency effects in the sham condition do not differ for trials preceded by in-phase as compared to anti-phase stimulation; t(40) = 0.4, p = 0.7, arguing against systematic offline influence of tACS on behavioral congruency. ”

2) Approach/Results. Authors should provide evidence that the key results are not contaminated by potential carry-over effects from preceding blocks

We consider it unlikely that the results we report are contaminated by carry-over effects for several reasons (see response to comment #1), most notably that we do not find any significant differences in sham periods following either in phase or anti phase stimulation. Additionally, stimulation conditions were varied pseudo-randomly between stimulus blocks, further averaging out any potential systematic carry-over effects.

We have added a control analysis, testing whether behavioral congruency effects for the sham session depend on the condition of the preceding stimulus block, and further clarified the pseudo-randomised structure of the stimulation protocol.

“Post-hoc analysis indeed revealed that behavioral congruency effects in the sham condition do not differ for trials preceded by in-phase as compared to anti-phase stimulation; t(40) = 0.4, p = 0.7, arguing against systematic offline influence of tACS on behavioral congruency.”

“Stimulation conditions were alternated between blocks in pseudo-random fashion with the constraint that stimulation condition could never be repeated for two consecutive blocks.”

3) Results. Figure 2D raises the possibility that the anti-phase condition produces a correlation between aPFC BOLD and the congruency effect that is roughly equal and opposite of what is observed in the in-phase condition (as opposed to producing no correlation, for instance). If this is a reasonable interpretation, then this potentially intriguing effect should be noted in the Results and noted in the Discussion.

This interpretation matches our a-priori formulated hypothesis on the effect of anti-phase stimulation (see also main point 5). That hypothesis is grounded on the observation that stronger power in the theta-band leads to stronger neuronal inhibition and stronger negative correlation with BOLD (Scheeringa et al., 2011). Accordingly, in the anti-phase condition, stronger theta-entrainment in aPFC (indexed by reduced tACS-related BOLD) would lead to stronger desynchronization between aPFC and SMC, and thus larger congruency effects. As indicated by the reviewers, the data in Figure 2D are consistent with this hypothesis. However, the correlation between tACS-dose in aPFC and behavioural congruency effects was significant in the in-phase condition (Spearman’s Rho = 0.32, p = 0.04) but not in the anti-phase condition (Spearman’s Rho = –0.23, p = 0.14). Figure 2—figure supplement 1E suggests that the weaker effects of anti-phase tACS on behavior might be due to enhanced activity in localized regions contralateral to the prefrontal stimulation. Namely, anti-phase stimulation (as compared to in-phase) enhanced BOLD responses in anatomically connected nodes of the emotional control circuit (aPFC, PPC (Mars et al., 2011; Koch et al., 2018)). Speculatively, this enhanced BOLD response might be interpreted as the emotional control circuit partially compensating for the aPFC-SMC desynchronization caused by the anti-phase stimulation. This compensation might lead to a weaker tACS-behavior relation in the anti-phase condition. However, post-hoc exploration did not reveal a straightforward relationship between local effects (in left aPFC, PPC) and behavioral congruency in the anti-phase condition. A dedicated study might test whether network-level effects explain the behavioral consequences of those enhanced responses in the emotional control circuit following anti-phase stimulation. We have added these considerations and added post-hoc analyses to the manuscript:

“We did not observe the anticipated anti-phase tACS induced increase of congruency effects caused by a desynchronization of aPFC and SMC (Polanía et al., 2015). […] Future studies could focus on characterizing these potential compensatory effects.”

“In the anti-phase condition, the correlation between tACS-dose and behavioural congruency effects was not significant (Spearman’s Rho = –0.23, *p* = 0.14). […] However, the BOLD signal difference between anti-phase and in-phase conditions extracted from aPFC and PPC (Figure 2—figure supplement 1E) did not correlate with anti-phase behavioural congruency effects (both r(39) < 0.14 , p > 0.3)”.

4) Results. In general, the paper should be revised to clearly indicate the polarities and valence of tACS stimulation, BOLD effects, and changes in congruency effect.• What is the difference between negative tACS dose in-phase and positive tACS dose anti-phase stimulation? The scatter plot and regression line in Figure 2D do not clearly communicate the difference between in-phase and anti-phase tACS with complementary tACS dose.

Polarities of tACS stimulation: The polarity of a tACS intervention changes sinusoidally over time within each cycle. We have clarified this:

“Stimulation was applied on-line during task performance in blocks of approximately 60 seconds (i.e. the length of a stimulus block; 12 trials), and consisted of electrical fields that changed polarity between the inner- and outer rings in theta-band (6 Hz) frequency over the aPFC and gamma-band (75 Hz), tapered with a 6 Hz sine wave over SMC”

Valence of tACS stimulation: It is not clear to us what the reviewers mean by “valence” of the tACS stimulation. Assuming the issue is about the phase relation between the two stimulation sites, then this information is provided in Figure 2A, B and in the subsection “tACS stimulation parameters”.

BOLD effects: We considered two independent BOLD effects. First, we analysed BOLD signals to verify that emotional congruency demands evoked activity in brain locations (notably aPFC) and with effect sizes matching previous reports (Figure 1). Second, we used BOLD signal analyses to quantify the effect of tACS on aPFC. To provide an unbiased estimate of how strongly aPFC is perturbed by stimulation, we computed a tACS-dose estimate, i.e. the degree to which aPFC BOLD response differs between stimulation (both in-phase and anti-phase conditions combined) and sham. This tACS-dose estimate is a combination of potential entrainment and compensatory effects acting on the right aPFC under the electrodes. We have tried to clarify this reasoning,

“Overall stimulation effects were estimated by comparing in-phase and anti-phase stimulation conditions to the sham condition, aggregated across congruent and incongruent conditions. We reasoned that this contrast would combine potential entrainment and compensatory mechanisms and thereby result in an unbiased estimate reflecting the extent to which the underlying neural tissue was perturbed by the stimulation; yielding a “tACS-dose” measure.”

Changes in congruency effect: The tACS-related changes in behavioural congruency effects are reported in Figures 2 and 3. The behavioral congruency effect was illustrated in Figure 1, as the difference in percentage correct between the incongruent and congruent conditions. Following the reviewers’ comments, we now introduce the congruency effect (Figure 1B) as error rate, i.e. in a manner consistent with its use in Figures 2, 3.

The scatter plot and regression line in Figure 2D do not clearly communicate the difference between in-phase and anti-phase tACS with complementary tACS dose: The interpretation of the relationship between behavioral congruency in each condition, and the BOLD response depends on the negative relationship between theta-band activity and BOLD effects (Scheeringa et al., 2011). Accordingly, reduced BOLD during theta-band stimulation is taken as an indication of stronger entrainment of the endogenous theta-band activity. This interpretational framework yields contrasting predictions for the two in-phase and anti-phase conditions. In the in-phase condition, increased aPFC theta-band entrainment would lead to reduced congruency effects, due to increased phase-amplitude coupling between aPFC and SMC. This situation would result in a positive correlation between tACS dose and behavioral congruency effects. In the anti-phase condition, increased entrainment would bias aPFC theta phase away from SMC dynamics. This situation would lead to larger congruency effects, i.e. a negative correlation between tACS dose and behavior. This is the pattern we observe (although not significantly for the anti-phase condition).

We could not find a better way to graphically present individual datapoints as well as summary statistics than Figure 2D. However, we have clarified the figure legend describing panel 2D as follows:

“D) Participants with stronger inhibitory responses to theta-band stimulation over aPFC (manifested as decreased BOLD signal; Scheeringa et al., 2011) improved their control over emotional actions (decreased congruency effect) during aPFC-SMC in-phase tACS (in green), but not during aPFC-SMC anti-phase tACS (in red), reflected in the interaction between Emotional Control (congruent, incongruent), Stimulations Phase (in-phase, anti-phase) and tACS-Dose (BOLD signal in aPFC during stimulation vs. sham (panel C)); F(1,39) = 9.3, p = 0.004, partial eta^2^ = 0.19. […] The direction of the effect in the anti-phase condition tentatively suggests that stronger entrainment to the theta-band stimulation (decreased BOLD) increases congruency effects, as would be expected when aPFC-SMC communication is disrupted.”

5) Discussion. Null results, and potential implications for major take-home points. Authors need to provide some discussion of their unsupported predictions.• In their preregistered hypothesis (Open Science Framework), the authors predict an increase of congruency effect in the anti-phase stimulation condition (subsection “Behavioural analyses”, last paragraph), however, this expectation was not confirmed by the results (subsection “Dose dependent effects of tACS on social-emotional control”, second paragraph). The absence of the effect of anti-phase tACS that was expected to be in the opposite direction to the effect of in-phase stimulation raises questions about the causal involvement of aPFC-SMC coupling in emotional-action control.• It is clear from the results that strengthening aPFC-SMC coupling enhances control.• However, if the disruption of such in-phase coupling by the anti-phase stimulation doesn't lead to the reduction in control it suggests that some other mechanisms might play a compensatory role. I recommend addressing this issue in the discussion of the results.

This issue largely overlaps with point #3, and we refer the reviewers to the arguments put forward there to interpret the unsupported prediction of the effect of anti-phase tACS.

We have amended the Discussion section to more clearly emphasize these points (see also point #3):

“We did not observe the anticipated anti-phase tACS induced increase of congruency effects caused by a desynchronization of aPFC and SMC (Polanía et al., 2015). […] Future studies could focus on characterizing these potential compensatory effects.”

6) Discussion/Abstract. The authors emphasize the clinical implications, but this discussion should be qualified and softened.• In this experiment, a substantial change in brain activity (as measured by BOLD) leads to a reliable, but relatively subtle behavioral change in a task that asks participants to override a mild response tendency. Indeed, according to Figure 1B, even subjects performing relatively poorly in the incongruent condition are still getting it right more than 80% of the time, suggesting that this condition presents no more than a mild challenge.• Emotional control will inevitably be harder in a real world scenario in which individuals will certainly perceive there to be far more at stake – this will be doubly true for any clinical population.• Thus, claims like "Similar interventions could help patients suffering from emotional-social disorders" and others should be softened. The authors present fascinating research that should be characterized more as a first step in developing gain-of-function therapeutics, instead of a treatment ready for, or even on the cusp of, application in the clinic.

We agree with the reviewers that our claims were a bit strong and the manipulation needs further study before clinical applications are possible. We have amended the Discussion section accordingly:

“These findings provide a first stepping stone towards the development of interventions aimed to help patients suffering from social-emotional disorders. […] Future studies could explore whether effects of in-phase stimulation hold for more challenging emotional control situations (e.g. Kaldewaij et al., 2019) and generate long-lasting effects through exposure treatment.”

References:

Mars RB, Jbabdi S, Sallet J, O’Reilly JX, Croxson PL, Olivier E, Noonan MP, Bergmann C, Mitchell AS, Baxter MG (2011) Diffusion-weighted imaging tractography-based parcellation of the human parietal cortex and comparison with human and macaque resting-state functional connectivity. J Neurosci 31:4087–4100.

Neuling T, Rach S, Herrmann CS (2013) Orchestrating neuronal networks: sustained after-effects of transcranial alternating current stimulation depend upon brain states. Front Hum Neurosci 7:161.

Vosskuhl J, Strüber D, Herrmann CS (2018) Non-invasive Brain Stimulation : A Paradigm Shift in Understanding Brain Oscillations. 12:1–19.